# Icariin Protects H9c2 Rat Cardiomyoblasts from Doxorubicin-Induced Cardiotoxicity: Role of Caveolin-1 Upregulation and Enhanced Autophagic Response

**DOI:** 10.3390/nu13114070

**Published:** 2021-11-14

**Authors:** Miriam Scicchitano, Cristina Carresi, Saverio Nucera, Stefano Ruga, Jessica Maiuolo, Roberta Macrì, Federica Scarano, Francesca Bosco, Rocco Mollace, Antonio Cardamone, Anna Rita Coppoletta, Lorenza Guarnieri, Maria Caterina Zito, Irene Bava, Luca Cariati, Marta Greco, Daniela Patrizia Foti, Ernesto Palma, Micaela Gliozzi, Vincenzo Musolino, Vincenzo Mollace

**Affiliations:** 1Institute of Research for Food Safety and Health (IRC-FSH), Department of Health Sciences, University “Magna Graecia” of Catanzaro, 88100 Catanzaro, Italy; miriam.scicchitano@hotmail.it (M.S.); saverio.nucera@hotmail.it (S.N.); rugast1@gmail.com (S.R.); jessicamaiuolo@virgilio.it (J.M.); robertamacri85@gmail.com (R.M.); federicascar87@gmail.com (F.S.); boscofrancesca.bf@libero.it (F.B.); rocco.mollace@gmail.com (R.M.); tony.c@outlook.it (A.C.); annarita.coppoletta@studenti.unicz.it (A.R.C.); lorenzacz808@gmail.com (L.G.); mariacaterina.zito@libero.it (M.C.Z.); irenebava@libero.it (I.B.); lucacariati1987@gmail.com (L.C.); palma@unicz.it (E.P.); micaela.gliozzi@gmail.com (M.G.); v.musolino@unicz.it (V.M.); mollace@libero.it (V.M.); 2Department of Health Sciences, University “Magna Graecia” of Catanzaro, 88100 Catanzaro, Italy; marta.greco@unicz.it; 3Department of Experimental, Clinical Medicine University “Magna Graecia” of Catanzaro, 88100 Catanzaro, Italy; foti@unicz.it; 4IRCCS San Raffaele Pisana, 88163 Roma, Italy

**Keywords:** cardiomyoblasts, Doxorubicin, Icariin, oxidative stress, autophagy

## Abstract

Doxorubicin (Doxo) is a widely used antineoplastic drug which often induces cardiomyopathy, leading to congestive heart failure through the intramyocardial production of reactive oxygen species (ROS). Icariin (Ica) is a flavonoid isolated from Epimedii Herba (Berberidaceae). Some reports on the pharmacological activity of Ica explained its antioxidant and cardioprotective effects. The aim of our study was to assess the protective activities of Ica against Doxo-detrimental effects on rat heart-tissue derived *embryonic cardiac* myoblasts (H9c2 cells) and to identify, at least in part, the molecular mechanisms involved. Our results showed that pretreatment of H9c2 cells with 1 μM and 5 μM of Ica, prior to Doxo exposure, resulted in an improvement in cell viability, a reduction in ROS generation, the prevention of mitochondrial dysfunction and mPTP opening. Furthermore, for the first time, we identified one feasible molecular mechanism through which Ica could exerts its cardioprotective effects. Indeed, our data showed a significant reduction in Caveolin-1(Cav-1) expression levels and a specific inhibitory effect on phosphodiesterase 5 (PDE5a) activity, improving mitochondrial function compared to Doxo-treated cells. Besides, Ica significantly prevented apoptotic cell death and downregulated the main pro-autophagic marker Beclin-1 and LC3 lipidation rate, restoring physiological levels of activation of the protective autophagic process. These results suggest that Ica might have beneficial cardioprotective effects in attenuating cardiotoxicity in patients requiring anthracycline chemotherapy through the inhibition of oxidative stress and, in particular, through the modulation of Cav-1 expression levels and the involvement of PDE5a activity, thereby leading to cardiac cell survival.

## 1. Introduction

According to the World Health Organization, cardiovascular diseases (CVDs) account for about one third of all global deaths. Among all cardiovascular deaths, cancer- related heart diseases are serious and dramatic health problems. Indeed, the antineoplastic therapy administered to improve cancer patient survival often leads to the onset of CVDs [1]. In particular, CVDs, such as heart failure (HF) and myocardial dysfunction, account for about half of the deaths of cancer patients [2,3]. Among the antineoplastic drugs capable of inducing severe late side effects, anthracyclines are a well-known example [4].

Doxorubicin (Doxo), also known as Adriamicyn, is an anthracycline isolated from the bacterium *Streptomyces peucetius* var. *caesius*, and is the hydroxylated congener of daunorubicin [5].

As a broad-spectrum chemotherapy agent, Doxo is routinely used in the treatment of several cancers, including breast, lung, gastric, ovarian, thyroid, non-Hodgkin’s and Hodgkin’s lymphoma, multiple myeloma, sarcoma, and, in particular, in pediatric cancers [6]. However, the use of Doxo is limited by cumulative dose-related cardiotoxic side effects that often lead to irreversible cardiomyopathy and congestive HF [7]. 

Currently, the mechanisms responsible for the onset of cardiotoxicity are not completely understood. Nevertheless, oxidative stress represents the main and best described mechanism proposed to explain Doxo-induced cardiotoxicity [8]. 

The cytostatic activity of Doxo has been observed to result in the massive production of reactive oxygen species (ROS), which contribute to cytotoxicity. ROS affect cardiomyocytes more than cancer cells due to their dependence on oxidative substrate metabolism compared to the glycolytic metabolism of cancer cells [9]. 

The production of ROS is mainly due to the Doxo redox cycle. Notably, Doxo binds cardiolipin and accumulates within the mitochondria. Here, NADH dehydrogenase in the mitochondrial respiratory complex I reduces Doxo, resulting in the production of ROS, such as superoxide (O_2_^−^) and hydrogen peroxide (H_2_O_2_), and electron transport chain (ETC) dysfunction by removing electrons normally used for the production of ATP [10]. As a result, reduced energy production, oxidation of mitochondrial DNA and loss of mitochondrial membrane potential are triggered, leading to the generation of mitochondrial permeability transition pore (mPTP) [11,12]. 

mPTP exists in two opening patterns: low conductance transient openings, which contribute to Ca^2+^ homeostasis, or long-lasting openings, which contribute to the rapid passage of ions and large molecules resulting in cell death [13].

Among the signals that regulate the opening of mPTP, the main mediators are the excessive mitochondrial concentration of Ca^2+^, the depolarization of ΔΨm and the presence of oxidative stress, which lead to a massive and non-selective mPTP opening [13,14]. 

ROS and reactive nitrogen species (RNS) mediated opening of mPTP is due to direct modulation of the pore or electron transport chain (ETC). In particular, transient pore opening has been shown to be associated with a transient depolarization of ΔΨm, linked to a sudden burst of O_2_^−^ generation related to ETC activity and initiation of local ROS signalling [15].

Therefore, Doxo-induced mPTP opening due to exacerbation of oxidative stress leads to mitochondrial dysfunction and depolarization, respiratory chain uncoupling, inhibition of ATP production, Ca^2+^ efflux, matrix swelling, dissipation of ΔΨm, and an increase in permeability which, in turn, results in the release of small pro-apoptotic proteins impairing cardiac energetics and damaging cardiomyocytes [16,17]. Finally, the increase in apoptotic and necrotic cell death is accompanied by the alteration of the contractile performance of the myocardium [18], which leads to myocardial dysfunction and HF [18,19].

Caveolins are the structural protein components of caveolar membranes, and they play different roles in the regulation of endothelial function, cellular lipid homeostasis and cardiac function by affecting nitric oxide (NO) activity and biogenesis [20]. Modulation of Caveolin-1 (Cav-1) expression levels has been observed to significantly affect apoptosis [21,22], while the scientific evidence that Cav-1 modulation affects Doxo-induced apoptosis is less known.

Apoptosis is one of the main mechanisms by which Doxo induces cardiomyocyte death [23,24].

Moreover, Doxo treatment has been found to cause an initial activation of the autophagic pathway, able to counteract Doxo-induced cardiotoxicity. In this condition, ROS levels exacerbate the upregulation of autophagic markers increasing the ratio of LC3II/LC3I and the level of Beclin-1 [25]. Initial activation of the autophagic process in response to Doxo damage acts as a protective mechanism; however, excess oxidative stress inhibits lysosome degradation resulting in cell death. In this condition, a disruption of normal protein degradation occurs in cardiomyocytes, and the subsequent increase in ubiquitinated proteins leads to the accumulation of autophagosomes and dysregulation of autophagic flux [26,27]. 

Several studies suggest that the administration of bioactive molecules together with anti-cancer drugs may play a beneficial role in cardiovascular protection due to their pleiotropic antioxidant effect [11,28,29,30]. Flavonoids, for example, thanks to their radical-scavenging activity, can be considered highly protective molecules against chronic toxicity caused by Doxo [31,32,33,34].

Icariin (Ica) is a flavonoid with antioxidant activity isolated from several species of Epimedii Herba (Berberidaceae,) such as E. *brevicornum Maxim*, E. *sagittatum* (Sieb. et Zucc.) *Maxim*, and E. *pubescens Maxim* [35,36], a plant widespread in the Mediterrànean regions and belonging to the plants used in traditional medicine for their tonic properties and aphrodisiac effects [37].

Several studies on the pharmacological activity of Ica identified important anti-cancer and anti-inflammatory effects and have improved its anti-oxidant and cardioprotective properties [38,39,40,41,42]. 

In particular, Ica decreases cardiac apoptosis induced by oxidative stress through mitochondrial protection, preserving Ca^2+^ homeostasis and increasing antioxidant enzymatic activity. [43]. In addition, Ica pre-treatment leads to reduced expression levels of caspase 3 [43], the down-regulation of Bax and caspase-9, and the upregulation of Bcl2 in cardiomyocytes [44,45].

In several experimental animal models it has been shown that Ica treatment improves isoproterenol-induced cardiac dysfunction and remodeling [44], exerts an anti-antiarrhythmic effect [46], reduces infarct size after myocardial ischemia-reperfusion (I/R) injury [47,48], and prevents pro-inflammatory pathway activation and ROS production in isoproterenol-induced takotsubo rat models [49].

Interestingly, Ica exhibits a specific inhibitory effect against phosphodiesterase-5a (PDE5a), suggesting a molecular mechanism underlying its beneficial effect on the cardiovascular system. Indeed, PDE5a inhibitors could be suitable for the treatment of cardiovascular diseases, thanks to their mechanism of action, which involves NO signalling [50,51,52].

PDE5a inhibitors could also be useful in counteracting Doxo-induced cardiotoxicity. Indeed, Doxo-induced cardiac apoptosis was significantly prevented by the well-known PDE5a inhibitor Sildenafil, which increases antioxidant enzyme levels, preserves Δψm, myofibrillar integrity preventing LV dysfunction without affecting the antineoplastic effect of Doxo [53,54].

On this basis, in our study, Ica isolated from E. *sagittatum*, was tested to assess its protective activities against Doxo-induced detrimental effects on the myocardium and to identify, at least in part, the potential molecular mechanisms capable of counteracting Doxo cardiotoxicity on rat H9c2 cardiomyoblasts. 

## 2. Materials and Methods

### 2.1. Cell Culture 

The study was conducted on an adherent H9c2 line of rat embryonic cardiomyoblasts (ATCC, Rockville, MD, USA). The cells were grown in Dulbecco’s modified Eagle’s medium (DMEM) reinforced with 10% fetal bovine serum (FBS), 100 U/mL penicillin, 100 µg/mL streptomycin, into a humidified 5% CO_2_ atmosphere at 37 °C. The medium was changed every two to three days and the cells were cultured to 70–80% confluence in 100 mm dishes. Cells were trypsinized and disseminated for Ica treatments.

Ica at different concentrations was obtained by serial dilutions in DMSO and was subsequently filtrated. 

The cells were treated with different concentrations (1 µM, 5 µM, 10 µM, 20 µM) of Ica (Sigma Aldrich # I1286) and incubated for 24 h to evaluate the related viability.Increasing concentrations of Ica (1 µM, 5 µM, 10 µM, 20 µM) were used to pretreat H9c2 for 3 h. Subsequently, they were exposed to 1 µM Doxo (Sigma Aldrich # D1515) for an additional 24 h to assess relative viability.Ica concentrations of 1 µM and 5 µM were used to pretreat H9c2 for 3 h, which were subsequently exposed to 1 µM Doxo for an additional 24 h to evaluate ROS production. The same experimental design was used to evaluate cell viability, immunofluorescence staining and protein expression levels.

### 2.2. MTT Assay 

The MTT assay relies on the observation that live cells with active mitochondria reduce the 3-(4,5-dimethylthiazol-2-yl)-2,5-diphenyltetrazolium bromide (MTT) to a visible dark-blue formazan reaction product and provides an indirect measurement of cell viability. The cytotoxic/cytostatic effects of Doxo have been assessed by MTT assay (Sigma Aldrich #M2003, Saint Louis, MO, US). The H9c2 were disseminated in 96-well microplates at a density of 6 × 10^3^ and, after 24 h, were pretreated with Ica (final concentration 1 μM, 5 μM, 10 μM, 20 μM) for 3 h and for an additional 24 h with Doxo 1 μM. At the end of the treatment, the medium was replaced with phenol red free medium containing MTT solution (0.5 mg/mL) and, after 4 h of incubation, 100 μL of a 10% SDS solution was added to each well to solubilize the formazan crystals. The plates were gently mixed and the optical density was measured at 570 nm with a microplate spectrophotometer reader. The values were used to calculate cell viability.

### 2.3. Intracellular ROS Detection 

Intracellular ROS levels were examined using 2’,7’dichlorodihydrofluorescein diacetate (H2DCFDA, D6883 Sigma-Aldrich, 3050 Spruce Street, Saint Louis, MO 63103, USA), a cell-permeable non-fluorescent probe. After cleavage of the acetate groups by intracellular esterases and oxidation, the H2DCFDA is converted into the highly fluorescent 2’,7’-dichlorofluorescein (DCF). Cells were plated in a 6-multiwell at a density of 150 × 10^3^ and, after 24 h, incubated with Ica 1 μM or 5 μM alone, or with Ica and Doxo as described above. At the end of the experimental period, the cells were washed with PBS, fixed in 4% formalin and incubated with 4’,6-diamidino-2-phenylindole dihydrochloride (DAPI, D8417, Sigma-Aldrich, 3050 Spruce Street, Saint Louis, MO 63103, USA) for 20 min at room temperature to stain nucleic acids. Images were collected using a 40× objective under a confocal microscope. Image quantification was performed counting ~1000 cardiomyoblasts/slide for a minimum of three slides per sample, and the positive ROS fraction was expressed as a percentage of total cardiomyoblasts. 

### 2.4. Apoptosis

Apoptosis was evaluated by flow cytometry with the Alexa Fluor^®^ 488 annexin V/Dead Cell Apoptosis Kit (V13241, Invitrogen, 29851 Willow Creek Road Eugene, OR 97402, USA), which provides a fast and affordable test for apoptosis. The cells were disseminated in a 60 mm dish at a density of 6 × 10^5^ and treated as described above. Cells were washed with ice-cold PBS and suspended in 100 µL 1× annexin-binding buffer at cell density of 10^6^ 5 μL of Alexa Fluor^®^ 488 Annexin V and 1 μL of the propidium iodide (PI) solution were added to each 100 μL cell suspension and incubated at 37 °C in an atmosphere of 5% CO_2_ for 15 min. At the end of the incubation period, 400 μL of 1× annexin-binding buffer was added and the samples were analysed by flow cytometry. After staining a cell population with Alexa Fluor^®^ 488 annexin V and PI in the binding buffer provided, apoptotic cells will show green fluorescence, dead cells will show red and green fluorescence, and living cells will show little or no fluorescence. These populations can be easily distinguished using a flow cytometer (FACSCanto^TM^, BD Biosciences) with the 488 nm line of an argon ion laser for excitation. The apoptotic index was calculated as the mean percentage of apoptotic cells.

### 2.5. Mitochondrial Permeability Transition Pore

The Image-iT LIVE Mitochondrial Transition Pore Assay Kit (29851 Willow Creek Road, Eugene, OR 97402, USA) was used to detect the mitochondrial permeability transition pore mPTP). Cells are loaded with the acetoxymethyl ester of calcein dye, calcein AM, which passively diffuses into cells and accumulates in cytosolic compartments, including mitochondria. Once inside the cells, the intracellular esterases cleave the acetoxymethyl esters to release the polar fluorescent dye calcein, which does not cross the mitochondrial or plasma membranes in appreciable quantities in relatively short periods of time. The fluorescence of cytosolic calcein is quenched by the addition of CoCl_2_, while the fluorescence of mitochondrial calcein is maintained. Cells were plated on a coverslip in a 6-multiwell at a density of 150 × 10^3^ and, after 24 h, incubated with Ica 1 μM or 5 μM alone, or Ica and Doxo as described above. At the end of the experimental period, the cells were washed twice in PBS buffer and labeled with an appropriate amount of labeling solution (1.0 µL of each of the following solution: 1.0 mM calcein AM stock solution, 200 µM MitoTracker Red CMXRos stock solution, 1.0 mM Hoechst 33342 dye, and 1.0 M CoCl_2_) to cover cells. The cells were incubated for 15 min at 37 °C, protected from light. Cells were washed in warm buffer and mounted. Fluorescence was detected using a 40× objective under a confocal laser scanning microscope (Leica TCS SP5, Leica Microsystems, Wetzlar, Germany). Fluorescence was quantified using ImageJ^®^ software by converting intensity to a greyscale value based on the RGB color model as previously described [55]. 

### 2.6. MitoSOX

MitoSOX red (Thermo Fisher #M36008, Waltham, MA, USA) was used for direct measurements of superoxide generated in the mitochondria of living cells. MitoSOX™ Red reagent is permeable to live cells and is rapidly and selectively targeted to mitochondria. Once in the mitochondria, the MitoSOX™ Red reagent is oxidized by the superoxide and exhibits a red fluorescence. MitoSOX™ Red reagent is readily oxidized by superoxide but not by other ROS- or reactive nitrogen species (RNS)–generating systems, and oxidation of the probe is prevented by superoxide dismutase. The oxidation product becomes highly fluorescent upon binding to nucleic acids. Cells were plated on a coverslip in a 6-multiwell at a density of 150 × 10^3^ and, after 24 h, incubated with Ica 1 μM or 5 μM alone, or Ica and Doxo as described above. After treatment, the MitoSOX reagent was dissolved in dimethyl sulfoxide (5 mM), diluted to 5 µM in serum-free medium. Also, 1 µM/µL 4’,6-diamidino-2-phenylindole dihydrochloride (DAPI, D84171, Sigma-Aldrich, Saint Louis, MO, USA) was diluted in serum-free medium. A suitable quantity of this solution was then added to the cells followed by incubation for 10 min at 37 °C; the cells were then washed twice with PBS and mounted. Fluorescence was detected using a 40× objective under a confocal laser scanning microscope (Leica TCS SP5, Leica Microsystems, Wetzlar, Germany) and quantified using ImageJ^®^ software by converting intensity to a greyscale value based on the RGB color model as previously described [55]. 

### 2.7. Protein Extraction and SDS-PAGE Western Blot Analysis

The cells were disseminated in 100-mm plates at a density of 125 × 10^4^ and treated as described above. At the end of the experimental period, the cells were washed with ice-cold PBS and lysed with lysis buffer (containing 50 mM Tris-HCl pH 7.4, 0.1% SDS, a NaCl 150 mM, EDTA 1 mM, TritonX-100 2.5%, DTT 5 mM, and protease and phosphatase inhibitor cocktail). A total of 20 μL of the supernatant was used to determine the total protein concentration by Bradford assay (Quick Start Bradford 1X Dye reagent, Bio-Rad #500-0205, Hercules, CA, USA) using bovine serum albumin (Quick Start Bovine Serum Albumin Standard, Bio-Rad #500-0206, Hercules, CA, USA) as a standard. The proteins were heat-denatured for 5 min at 95 °C in the sample loading buffer (500 mM Tris/HCl, pH 6.8; 30% glycerol; 10% sodium dodecyl sulfate; 5% β-mercaptoethanol; and 0.024% bromophenol blue), and 20 μg protein lysate were resolved by sodium dodecyl sulfate polyacrylamide gel electrophoresis and transferred to nitrocellulose membranes (Amersham Protan 0.2 µm NC 10600001, Little Chalfont, UK). The membranes were then blocked with Tris/HCl (pH 7.6) containing 0.1% Tween 20 and 5% BSA for 1 h and incubated overnight at 4 °C shaking with the following solution containing the primary antibodies: anti-Caveolin-1 (Abcam #ab36152, Lot: GR3362124-2; Cambridge CB2 0AX, UK); Beclin-1 (Cell signalling #3738, Lot: 3; Danvers, MA, USA); Anti-LC3 (MBL PM036; Lot: 035, 15A Constitution Way, Woburn, MA 01801); anti-Actin (Sigma Aldrich #A3853, Lot: 048M4754V; #0 Saint Louis, USA); anti-Tubulin (Abcam #ab6046, Lot: GR3194813-1). The membranes were then washed in Tris-buffered saline (TBS, pH 7,6) with 0,1% Tween-20 and incubated with horseradish peroxidase-conjugated secondary antibodies (anti-rabbit antibody Pierce #31460 or anti-mouse antibody Pierce #31430, Invitrogen, Carlsbad, CA, USA) for 1 h at room temperature and shaking. Bound antibodies were visualized using the chemiluminescent kit (ECL WB Detection, GE Healthcare RPN210601819, Little Chalfont, United Kingdom). Immunoblots scanning and analyses were performed using an imaging system (UVITEC Imaging Systems, Alliance, Cambridge, United Kingdom). Bands quantification was performed using UVITEC Imaging Systems Alliance software (Cambridge, UK).

### 2.8. PDE5a Activity Assay

The PDE5a assay Kit (PDE5a Assay Kit, #60350, BPS Bioscience, San Diego, CA, USA) was used for the identification of PDE5a inhibitors by fluorescence polarization. The assay is based on the binding of a fluorescent nucleotide monophosphate generated by PDE5a to the binding agent. Briefly, phosphodiesterases catalyze the hydrolysis of the phosphodiester bond into dye-labeled cyclic monophosphates. The beads selectively bind the phosphate group in the nucleotide product. In the polarization assay, dye molecules with absorption transition vectors parallel to linearly polarized excitation light are selectively excited. Dyes attached to the rapidly-rotating cyclic monophosphates emit light with low polarization. Dyes attached to the slowly-spinning nucleotide-bead complexes emit highly polarized light. First, fluorescently labeled cGMP (substrate) is incubated with PDE5a for 1 h. Second, the binding agent is added to the reaction mix to produce a change in the fluorescent polarization which can be measured using a fluorescence reader equipped to measure the fluorescence polarization. A sample with PDE5a was used as a positive control, cGMP was used as substrate, Sildenafil (#12814, Sildenafil Citrate, Farmalabor, Canosa, Italy) and Ica 1 μM and 5 μM were used as inhibitors to compare their inhibitory activity. The substrate and binding agent were added to the cell lysates and the fluorescence was analysed in the presence or absence of Sildenafil.

### 2.9. Statistical Analysis

GraphPad PRISM 6.0 (GraphPad Software, Inc., La Jolla, CA, USA) was used to perform statistical analysis. The results are shown as mean ± SEM. Normality was tested using the D’Agostino Pearson’s test. Data which have Gaussian distribution were analysed by one-way ANOVA followed by Tukey’s test, while data without normal distribution were analysed using Kruskal–Wallis analysis of variance and subsequent Dunn’s tests or Mann-Whitney test. A *p*-value of <0.05 was considered significant.

## 3. Results

### 3.1. The Detrimental Effects of Doxo on Mitochondrial Metabolism and Cell Viability in H9c2 Cardiomyoblasts

The effect of Doxo treatment on H9c2 mitochondrial metabolism and cell viability was investigated. H9c2 cells were treated with Doxo 1 μM at 18 h, 24 h and 48 h.

At the end of the treatment, the MTT test was performed, showing a time-dependent reduction in mitochondrial metabolic activity correlated to a subsequent suppression of cell viability. After only 18 h, a slight reduction of the viability was observed (Figure 1). Interestingly, incubation of cardiomyoblasts with Doxo for 24 h leads to a significant reduction in mitochondrial metabolism, which is drastically reduced after 48 h (Figure 1). Therefore, subsequent experiments were performed focusing on 24 h of Doxo treatment, which best mimics a sub-toxic damage on cardiomyoblasts. 

### 3.2. The Protective Effects of Ica on Cell Viability in Doxo Treated H9c2 Cells

First, a dose-response curve was performed to verify the effects of Ica on Doxo-induced cardiotoxicity. Different concentrations of Ica (1 µM, 5 µM, 10 µM, 20 µM) were used 3 h prior to treatment with Doxo (1 μM) for 24 h. At the concentrations tested, Ica produced no toxic side effects (Figure 2, panel A). Data collected from the MTT test confirmed Doxo-induced cell toxicity. 

Interestingly, cell viability was significantly improved by Ica pretreatment at all concentrations tested. In particular, a significant improvement was registered starting from the lowest concentrations (Figure 2, panel B). Furthermore, an equally clear protection, exerted by Ica, was also found at higher concentrations, although not dose dependent (Figure 2, panel B). Since higher concentrations of Ica result in a plateau effect, while lower concentrations already significantly increase cell viability, subsequent experiments focused on the lowest concentrations capable of producing a significant beneficial effect (Ica 1 μM and 5 µM).

### 3.3. Ica Reduces Doxo-Induced ROS Overproduction

The measurement of intracellular ROS overproduction was performed by immunofluorescent assay based on the staining with the fluorescent probe H2DCFDA. Doxo treatment significantly increased ROS production compared to control cells, as observed by green stained positive cells. Pretreatment with Ica 1 μM and 5 µM for 3 h significantly reduced ROS levels at both concentrations. (Figure 3, panels A and B). 

### 3.4. Ica Inhibits Doxo-Induced O_2_^−^ Overproduction

To specifically determine mitochondrial O_2_^−^ overproduction, MitoSOX staining was performed. Treatment of H9c2 cells with Doxo alone significantly increased O_2_^−^ levels when compared to Ctrl cells, as observed by red-stained positive cells. Pretreatment with Ica 1 μM and 5 µM for 3 h significantly reduced O_2_^−^ levels at both concentrations compared to cells treated with Doxo alone (Figure 4, panels A and B).

### 3.5. Modulatory Effect of Ica on mPTP Opening in Doxo-Treated H9c2 Cells

To directly measure mitochondrial permeability transition pore opening, cells were loaded with acetoxymethyl ester of the calcein dye, calcein AM, while Mitotracker red staining was used to identify mitochondria. The fluorescence of cytosolic calcein was quenched by the addition of CoCl_2_. Treatment with Doxo resulted in a significant increase in the opening of mPTP. Interestingly, pretreatment for 3 h with both Ica concentrations significantly reduced mPTP opening, as showed by the increase in green fluorescence within the mitochondria compared to that of Doxo treated cells (Figure 5, panel A and B).

### 3.6. The Protective Effect of Ica against Doxo-Induced Apoptotic Cell Death 

Double staining for FITC-annexin V bond and cellular DNA using propidium iodide (PI) was performed. H9c2 cells incubated for 3 h with both Ica 1 μM and 5 μM did not lead to apoptotic or necrotic cell death (Figure 6, panel B). Conversely, treatment of cells with Doxo for 24 h dramatically led to apoptosis, as demonstrated by the increase from 10.2% (untreated cells) to 47.5% in the number of Annexin positive cells treated with Doxo alone (Figure 6, panel A). Pretreatment of cells with Ica significantly reduced the percentage of apoptotic cells at both concentrations considered (Figure 6, panel A).

### 3.7. The Role of Ica on Cav-1 Expression Levels in Doxo-Induced Cardiotoxicity 

To evaluate the effects of Ica on Cav-1 modulation in H9c2 cells, a western blot analysis was performed. In H9c2 cells, Doxo treatment was able to upregulate the expression levels of the protein Cav-1 (Figure 7, panels A and B) when compared to Ctrl cells. Interestingly, treatment with Ica at both concentrations significantly reduced the expression levels of the molecular marker considered (Figure 7, panels A and B).

### 3.8. Ica Inhibits PDE5a Activity in Doxo-Treated H9c2 Cells 

The PDE5A1 assay was used to identify PDE5a inhibitors using fluorescence polarization. The activity of PDE5a was observed by comparing the inhibitory effect of Sildenafil and Ica on the enzyme. Sildenafil has been observed to exert an important and significant inhibitory effect on PDE5a. Interestingly, Ica at a concentration of 1 µM tended to reduce PDE5a activity, while a significant inhibitory effect was observed using Ica 5 µM (Figure 8, panel A). This result confirms the concentration-dependent inhibitory effect of Ica compared with a well-known enzyme inhibitor. On cell lysates, Doxo treatment resulted in increased PDE5a activity compared to Ctrl or Ica alone at both concentrations. Conversely, pretreatment of cardiomyoblasts with Ica 1 µM or Ica 5 µM significantly reduced PDE5 activity when compared to Doxo-treated cells (Figure 8, panel B).

### 3.9. The Protective Role of Ica in the Autophagic Pathway in Doxo-Treated H9c2 Cells

Treatment of H9c2 cells with Doxo alone significantly upregulated the pro-autophagic marker Beclin-1 when compared to Ctrl cells (Figure 9A,B). Moreover, LC3I, and its autophagosome-associating form, LC3II, were studied. The measurement of LC3 in its cleaved form, derived from the ratio of LC3II/LC3I, was associated with the activation of autophagy. Interestingly, the LC3II/LC3I ratio was upregulated after incubating H9c2 cells with Doxo (Figure 9C,D). Conversely, pretreatment of Doxo-treated cells with 1 μM and 5 μM of Ica strongly restored Beclin-1 expression levels and LC3II/LC3I ratio compared to cells treated with Doxo alone (Figure 9A,C). 

## 4. Discussion

Our work clearly demonstrated the strong antioxidant properties and cardioprotective role of Ica in counteracting Doxo-induced ROS overproduction and mitochondrial dysfunction. Furthermore, we showed that this effect strongly prevents the activation of pro-apoptotic and pro-autophagic signalling pathways, avoiding the massive loss of cardiomyocytes. Finally, we identified, at least in part, one potential molecular mechanism by which Ica inhibits cardiac stress reactions and prevents Doxo cardiotoxicity. Indeed, in our experimental model, the modulation of Cav-1 protein expression levels together with the direct inhibition of the enzymatic activity of PDE5a are closely involved in the cardioprotective processes exerted by Ica.

Therefore, our data showed for the first time that Ica as a nutraceutic anti-oxidant compound has interesting cardioprotective effects in preventing or reducing Doxo-induced cardiotoxicity on H9c2 cardiomyoblasts.

Our data are in agreement with previous studies according to which the massive ROS stimulation induced by Doxo cytostatics results in cardiotoxicity, reducing cell viability [56,57].

Since the intracellular uptake of Doxo is time-dependent, the cellular mitochondrial metabolic activity and possible cytotoxic effect of Doxo on H9c2 cells were performed at three time points (18–24–48 h) [58,59].

Our results showed a significant reduction in cell viability after 24 h of treatment with 1 μM Doxo, which is the comparative concentration at the clinically relevant concentration capable of producing sub-lethal effects [10,60]. Interestingly, Doxo’s detrimental effect was prevented by pretreating H9c2 cells with increasing concentrations of Ica. In particular, a significant protective effect against Doxo cytotoxicity was already observed at the lowest Ica concentrations used (1 µM and 5 μM). 

Doxo is known to be responsible for the oxidative stress production through the interaction with different molecules, and can generate several types of ROS. Among all, NO can rapidly react with the O_2_^−^ produced in the mitochondria, generating the strong oxidizing peroxynitrite (ONOO^−^). and exceeding the O_2_^−^ scavenging activity of the SOD enzyme. Overall, these oxidants together with hydrogen peroxide (H_2_O_2_) represent the main components of Doxo-induced oxidative stress [58]. 

In the development of Doxo-induced cardiac damage, mitochondrial dysfunction represents the main cause of ROS production [61]; indeed, several studies demonstrated the central role of mitochondria in Doxo-induced cardiac injury [62,63]. Furthermore, Doxo is known to lead to the alteration of mitochondrial ETC through the activation of its “redox cycle” produced by NADH dehydrogenase. In particular, the production of O_2_^−^, H_2_O_2_ or OH. contributes to the impairment of ΔΨm [64], leading to a decrease in ATP production. As mitochondria represent the headquarters of heart energy, the latter mechanism contributes to inducing myocardial dysfunction [65].

To verify the role of oxidative stress in the development of Doxo-induced cytotoxicity and to determine the potential anti-oxidant activity of Ica against Doxo-induced ROS production, both intracellular and mitochondrial ROS levels were measured. In our experimental in vitro model, rat cardiomyoblasts showed a significant increase in ROS and reactive nitrogen species production after 24 h of treatment with Doxo, which was not observed in cells pretreated with 1 µM and 5 µM of Ica. Moreover, mitochondrial O_2_^−^ over-production was prevented by pretreating of H9c2 cells with both Ica concentrations considered.

These results confirm the anti-oxidant properties of Ica as previously reported [45], but also show the strong anti-oxidant potential of Ica in protecting cardiomyoblasts, in particular, against Doxo-induced ROS production and mitochondrial dysfunction.

As already observed in the literature, Doxo determines the alteration of the mitochondrial membrane also by cardiolipin interaction and alteration of the mPTP opening [66]. In the case of overproduction of free radical species, with a low conductance permeability transition pore flickering between open and closed states, the opening of mPTP is enhanced [67]. Indeed, ROS are able to directly modify the pore, resulting in the alteration of the transient depolarization of the mitochondrial membrane potential associated with a sudden burst of O_2_^−^ generation and initiation of local ROS signalling [15]. Based on these observations, we directly determined mPTP status. In the cardiomyoblasts pretreated with Ica, we detected “healthy” mitochondria when compared to the cells treated with Doxo alone which, instead, showed mPTP opening alteration and increased mitochondria permeability. These data are in agreement with evidence demonstrating the correlation between altered Ca^2+^ exchange, ROS overproduction, loss of ΔΨm and mPTP opening caused by Doxo [68,69,70], suggesting that Ica can represent a valid approach to counteract mitochondrial damage. Accordingly, some previous studies proposed that Ica treatment protects H9c2 cells from apoptosis by inhibiting endoplasmic reticulum stress through the reduction of ROS production. Moreover, the authors identified a significant reduction in the loss of ΔΨm and caspase 3 expression levels after Ica treatment [45,46].

Since Doxo-induced ROS overproduction represents a clear upstream inducer of different signalling pathways such as apoptosis, necrosis and autophagy processes, resulting in cardiomyocyte death [71], the activation of both intrinsic and extrinsic apoptotic pathways, as well as the necrotic pathway, is considered a central mechanism in the setting of cardiotoxicity [71,72]. Therefore, we also explored the role of apoptotic cell death. Our results confirmed that 24 h of Doxo treatment significantly increased apoptotic cell death when compared to Ctrl cells. Interestingly, the data also proved the significant protective effect of Ica against Doxo–triggered cell death at both considered concentrations. In addition, Ica alone did not result in any detectable evidence of apoptosis confirming its safety.

At this point, we wondered about the possible mechanism and the molecular targets underlying the cardioprotective effects of Ica.

As reported in the literature, Ica exerts a direct effect as a PDE5a inhibitor [73,74,75], which was further confirmed in our experiments. Indeed, we showed a PDE5a inhibitory activity of Ica 1 μM which becomes statistically significant at 5 μM when compared to the positive control and to the known PDE5a inhibitor sildenafil. In light of this result, we further investigated the modulation of PDE5a activity in the Doxo cardiotoxicity pattern, highlighting the possible protective effect exerted by Ica through direct inhibition of PDE5a.

In our in vitro experimental model, PDE5a activity was upregulated in Doxo-treated cells, while, intriguingly, reduced PDE5a activity was recorded in Ica pretreated cells. This finding is supported by the literature suggesting that the administration of a PDE5a inhibitor significantly reduces oxidative stress in the failing heart [76,77]. It is also known that NO production is closely related to PDE5a activity, as it is an important cGMP-degrading enzyme [78] Therefore, PDE5a inhibition, as in the case of cells pretreated with both 1 µM and 5 µM Ica, could restore the eNOS/iNOS rate, promoting beneficial and cardioprotective mechanisms against Doxo-induced cardiotoxicity [79,80]. Hence, our and other data suggest that the use of certain antioxidants and specific PDE5a inhibitors could be promising therapeutic approaches to attenuate myocardial oxidative stress.

It has recently been observed that apoptosis could be affected by Cav-1 [81]. Indeed, Cav-1 is involved in the regulation of several cell survival or cell death processes depending on the cell type [82,83,84]. Although the involvement of Cav-1 in Doxo-induced cardiotoxicity is still poorly investigated, an in vivo study showed that Cav-1 is required for Doxo-induced apoptosis. Conversely, the knockdown of Cav-1 prevents the activation of caspase 3 [22]. Furthermore, Doxo-induced p38 phosphorylation was inhibited after Cav-1 knockdown in an in vitro model of Doxo-induced cardiotoxicity, leading to a reduction in caspase-3 cleavage. Therefore, reduced Cav-1 expression appears to protect damaged cardiomyocytes but was probably insufficient to prevent overall Doxo-induced apoptosis because Cav-1 knockdown did not affect Doxo-induced ERK signaling [85]. In our hands, an increased expression of Cav-1 in Doxo-treated cardiomyoblasts was reported, while, intriguingly, this condition was significantly counteracted by pretreatment with both concentrations of Ica. Reduced Cav-1 expression levels are often associated with inhibition of the autophagic process [86,87,88,89].

It was observed that oxidative stress modulates the autophagic pathway since ROS can interfere with this mechanism at several levels. It was found that starved cells treated with hydrogen peroxide underwent a structural modification of Atg4 [90,91]. Furthermore, oxidative stress associated to Doxo promoted the autophagic process in cardiomyocytes, which could be protective or could induce damage [31,92].

As previously shown, the autophagic pathway is initially activated after Doxo treatment to counteract cardiotoxic damage. However, in this condition, ROS overproduction upregulates the pro-autophagic markers by increasing the LC3II/LC3I ratio and Beclin-1 levels [25].

Our data showed that Doxo-treatment activates the autophagic pathway in cardiomyoblasts as observed by the upregulation of Beclin-1 expression levels and LC3 lipidation.

Interestingly, for the first time, we demonstrated that the pretreatment of H9c2 cells with Ica at the lowest concentration (1 µM) significantly downregulates the autophagic pathway.

We hypothesize that the hyperactivation of the autophagic pathway triggered by the overproduction of free radicals can significantly contribute to the apoptotic cell death in Doxo-induced cardiotoxicity. The link between these two important pathways could be represented by the upregulation of the transcription factor p53. Indeed, as previously observed, Doxo can upregulate p53 expression levels, leading to the inhibition of mTOR [93]. Moreover, p53 mediates the suppression of the transcription factor GATA-4 and, therefore, could down-regulate the pro-survival protein Bcl-2 which, physiologically, prevents autophagy initiation by the binding with Beclin-1. In this context, Doxo could also promote Bcl-2 phosphorylation, leading to inhibition of the Bcl-2/Beclin-1 interaction and the facilitating of the initiation of autophagy [94].

Overall, our results focused on the strong antioxidant properties of Ica, highlighting a prominent cardioprotective activity against Doxo-induced cardiotoxicity. We also showed that these beneficial properties significantly prevent mitochondrial dysfunction and apoptotic cell death.

Interestingly, we identified for the first time one feasible molecular mechanism through which Ica could exert its cardioprotective effects by observing a significant reduction in the protein expression level of Cav-1, whose overexpression is closely related to the production of oxidative stress reactions and hyperactivation of pro-inflammatory and pro-apoptotic markers. Furthermore, we showed the ability of Ica to restore physiological levels of activation of the protective autophagic process, which is closely associated with Cav-1 modulation.

Finally, we recognized a specific direct Ica inhibitory effect on PDE5a activity, capable of improving mitochondrial function, thus suggesting that the use of certain antioxidants and specific PDE5a inhibitors could be a promising therapeutic approach to attenuate myocardial oxidative stress.

## 5. Conclusions

Our results suggest that Ica could be helpful in counteracting the side effects associated with anthracycline administration, through its cardioprotective activity. Thus, it might represent an important tool in the prevention of cardiac toxicity induced by Doxo.

## Figures and Tables

**Figure 1 nutrients-13-04070-f001:**
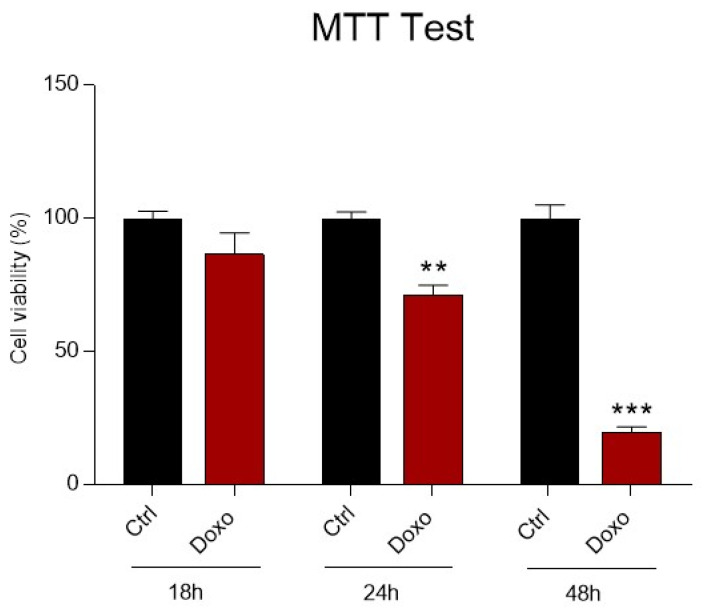
The effect of Doxorubicin treatment on H9c2 cell viability. H9c2 cells were treated with Doxorubicin (Doxo) 1 μM at three-time points (18 h, 24 h and 48 h). Treatment with Doxo induces a time dependent reduction in cell viability. Data are presented as mean  ±  SEM. ** *p* < 0.01 vs. Ctrl (24 h); *** *p* < 0.001 vs. Ctrl (48 h); Mann Whitney test (*n* = 8). Data is from three different independent experiments.

**Figure 2 nutrients-13-04070-f002:**
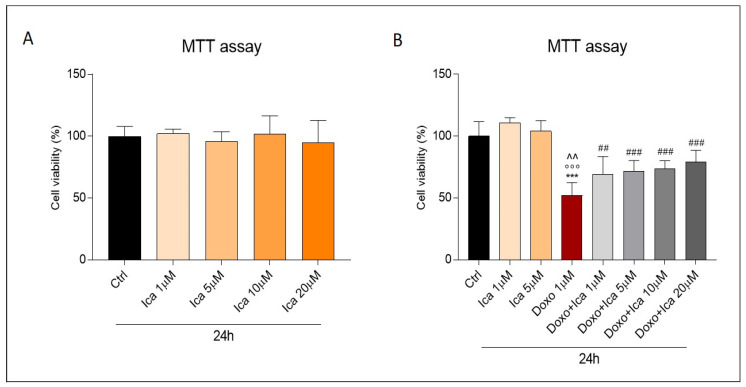
The effect of Icariin on Doxo-treated H9c2 cell viability. (**A**). Incubation of H9c2 cells with Icariin (Ica) alone, at increasing concentrations, did not result in a reduction of cell viability compared to the Ctrl group. (**B**). Pretreatment with increasing concentrations of Ica for 3 h protects cardiomyoblasts from Doxo induced toxicity. Data are expressed as mean ± SEM *** *p* < 0.001 vs. Ctrl; °°° *p* < 0.001 vs. Ica 1 μM; ^^ *p* < 0.01 vs. Ica 5 μM; ^##^
*p* < 0.01 vs. Doxo; ^###^
*p* < 0.001 vs. Doxo; Mann Whitney test (*n* = 8). Data is from three different independent experiments.

**Figure 3 nutrients-13-04070-f003:**
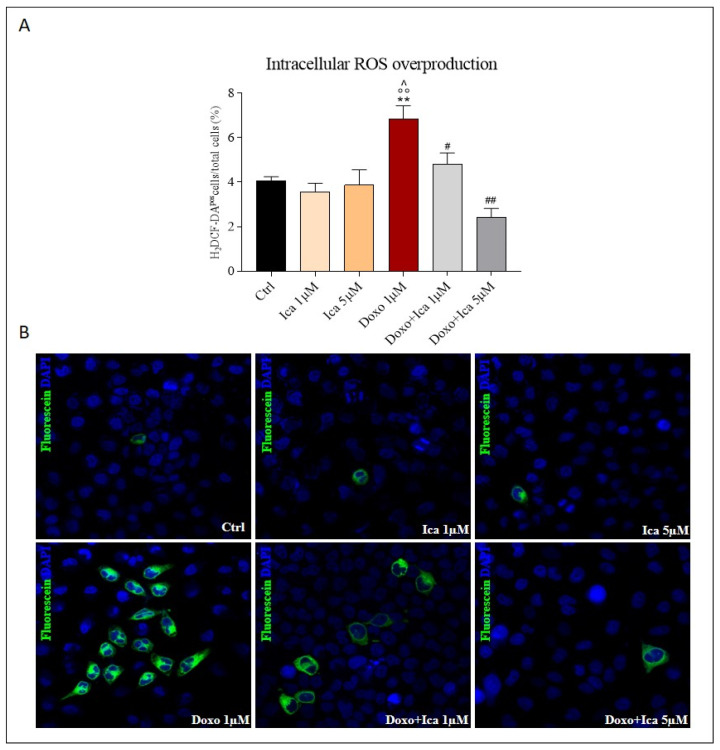
Ica reduces intracellular reactive oxygen species overproduction. (**A**). Reactive oxygen species (ROS) overproduction was determined by immunofluorescence assay. Oxidative stress was quantified by counting ∼1000 myoblasts/slide, for a minimum of three slides per sample, and the ROS positive fraction was expressed as a percentage of total myoblasts. (**B**). Representative confocal images of H9c2 cells treated as described above. Green staining showed fluorescent probe 2’,7’—dichlorodihydrofluoresceindiacetate (DCFH-DA), blue staining showed DAPI (4’,6-diamidino-2-phenylindole), used for nucleic acid staining. Cell images were collected using a 40× confocal microscope objective. Data are expressed as mean ± SEM, ** *p* < 0.01 vs. Ctrl; °° *p* < 0.01 vs. Ica 1 μM; ^^^
*p* < 0.05 vs. Ica 5 μM; # *p* < 0.05 vs. Doxo; ## *p* < 0.01 vs. Doxo; Mann Whitney test (*n* = 3). Data is for three independent experiments.

**Figure 4 nutrients-13-04070-f004:**
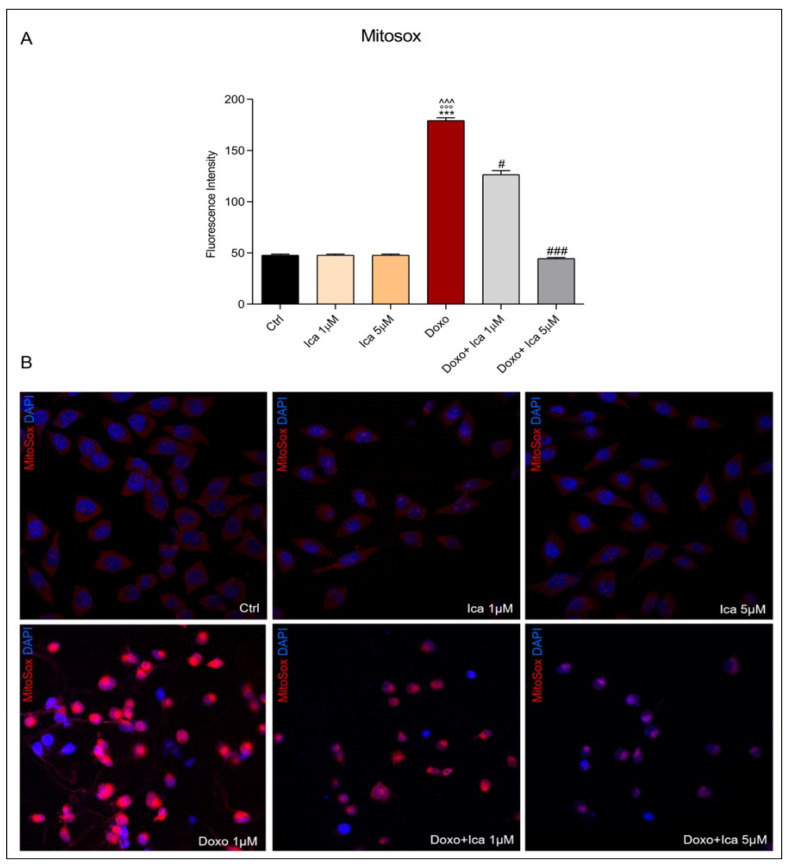
The effect of Ica on mitochondrial superoxide overproduction. (**A**). Superoxide (O_2_^−^) levels were determined by immunofluorescence assay. Fluorescence was quantified using ImageJ^®^ software by converting the intensity to a greyscale value based on the RGB colour model. For each treatment, 30–40 cells were processed. Significant differences in mean fluorescence were found between all treated cells. (**B**). Representative confocal images of H9c2 cells treated as described above. Red staining showed the MitoSOX fluorescent probe used to identify mitochondrial O_2_^−^ overproduction, blue staining showed DAPI (4’,6-diamidino-2-phenylindole), used for nucleic acid staining. Cell images were collected using a 40× confocal microscope objective. Data are expressed as mean ± SEM., *** *p* < 0.001 vs. Ctrl; °°° *p* < 0.001 vs. Ica 1 μM; ^^^ *p* < 0.001 vs. Ica 5 μM; # *p* < 0.05 vs. Doxo; ### *p* < 0.001 vs. Doxo; Kruskal-Wallis test and Dunn’s multiple comparisons test (*n*= 3). Data is for three independent experiments.

**Figure 5 nutrients-13-04070-f005:**
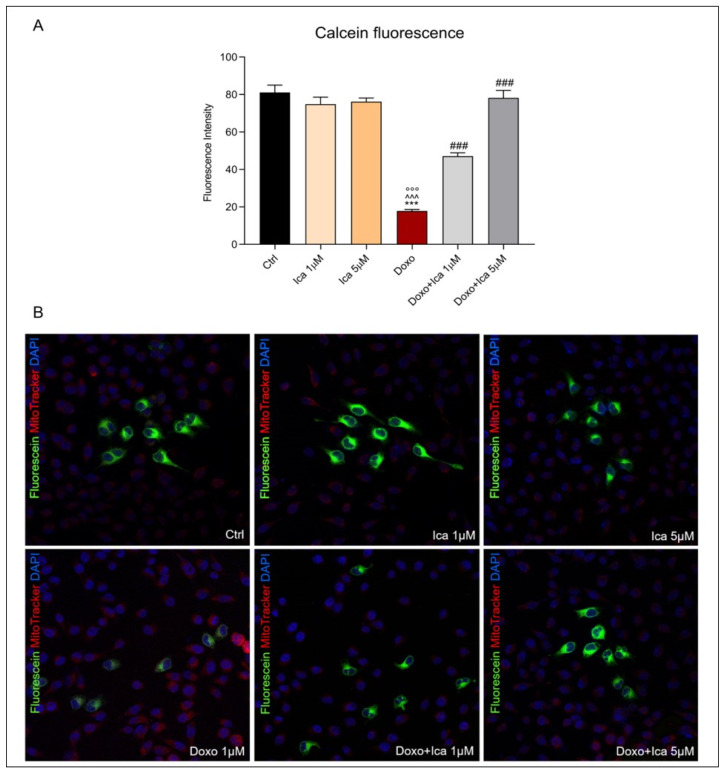
The effect of Ica on mitochondrial permeability transition pore opening in Doxo-treated H9c2 cells. (**A**). Mitochondrial permeability transition pore (mPTP) opening was determined by immunofluorescence assay. Fluorescence was quantified using ImageJ^®^ software by converting the intensity to a greyscale value based on the RGB color model. For each treatment, 30–40 cells were processed. Significant differences in mean fluorescence were found between all treated cells. (**B**). Confocal representative images of H9c2 cells treated as described above. Green staining showed calcein, blue staining showed Hoechst 33342 used for nucleic acid staining, red staining showed MitoTracker Red CMXRos used for mitochondria staining. Cell images were collected using a 40× confocal microscope objective. Data are expressed as mean ± SEM, *** *p* < 0.001 vs. Ctrl; °°° *p* < 0.001 vs. Ica 1 μM ^^^ *p* < 0.001 vs. Ica 5 μM; ### *p* < 0.001 vs. Doxo; Kruskal-Wallis test and Dunn’s multiple comparisons test (*n* = 3). Data is from three independent experiments.

**Figure 6 nutrients-13-04070-f006:**
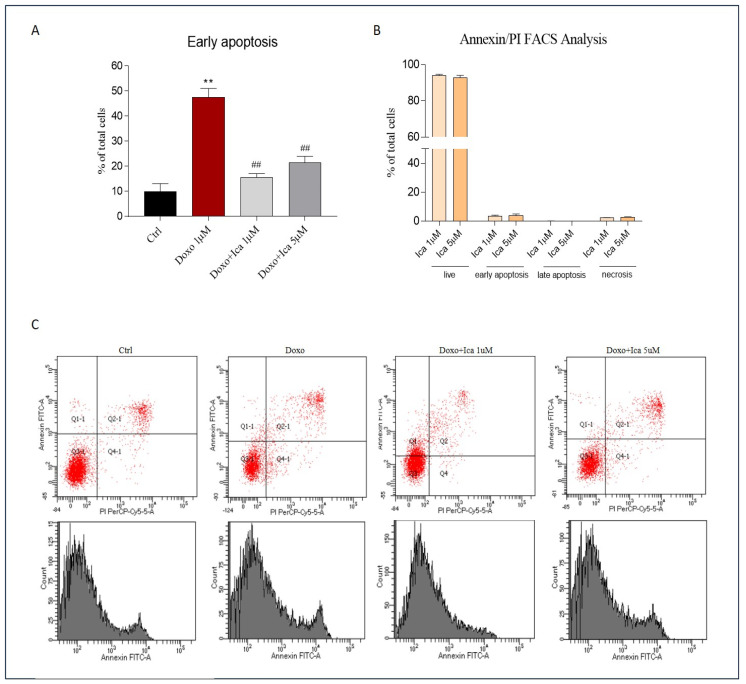
Cytofluorimetric evaluation of the effect of Ica against Doxo-induced apoptotic cell death. (**A**). H9c2 cells treated for 3 h with both Ica 1 μM and 5 μM showed no significant apoptotic or necrotic cell death. (**B**). Treatment with Doxo 1 μM for 24 h resulted in a detrimental increase in apoptotic cell death when compared to Ctrl cells. Conversely, pretreatment of H9c2 cells with Ica 1 μM and 5 μM significantly prevented apoptosis. (**C**) Representative cytofluorimetric dot plot. Data are expressed as means ± SEM.; ** *p* < 0.01 vs. Ctrl, ## *p* < 0.01 vs. Doxo; Mann Whitney test (*n* = 3). Data is for three independent experiments.

**Figure 7 nutrients-13-04070-f007:**
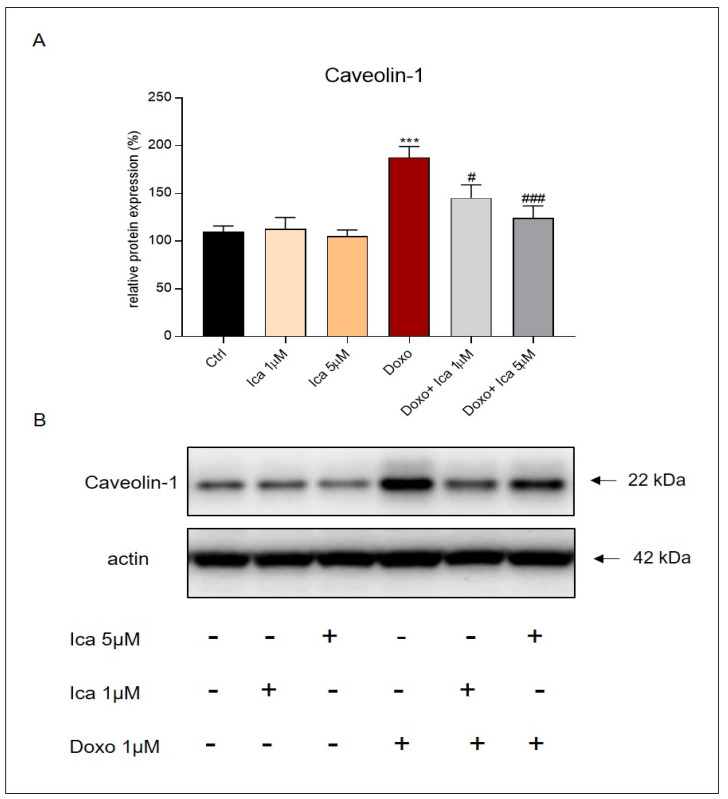
Ica inhibits Caveolin-1 expression levels in Doxo-treated H9c2 cells. (**A**) Doxo treatment significantly upregulated Caveolin-1 (Cav-1) expression levels compared to Ctrl cells. Pretreatment of Ica cells at both concentrations used significantly inhibited the expression of the molecular marker. (**B**) Representative images of western blot analysis. Three independent Western Blots were quantified by densitometry. The protein expression of Cav-1 was normalized with respect to the corresponding actin signals of the appropriate samples and expressed as a percentage. Data are expressed as mean ± S.E.M. *** *p* < 0.001 vs. Ctrl; # *p* < 0.05 vs. Doxo; ### *p* < 0.01 vs. Doxo; Mann Whitney test (*n* = 3).

**Figure 8 nutrients-13-04070-f008:**
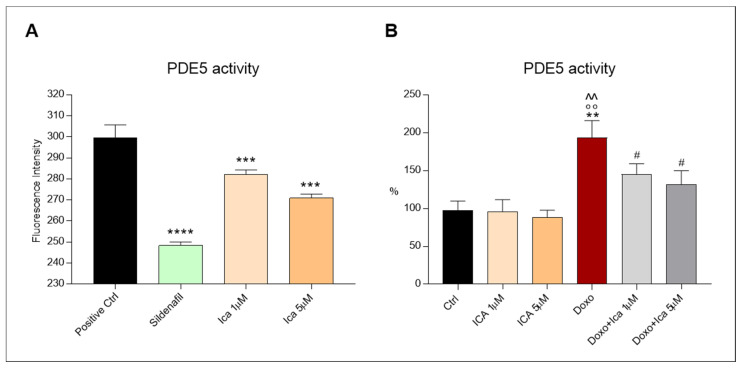
The inhibitory effect of Ica on phosphodiesterase-5a activity. (**A**). Treatment with Ica 1 µM resulted in a reduction of phosphodiesterase-5a (PDE5a) activity, while sildenafil and Ica 5 µM significantly reduced PDE5a activity. (**B**). On H9c2 cells, the increased activity of PDE5a, determined by Doxo, was significantly reduced by ICA1 µM and by ICA5 µM. Data are expressed as mean ± S.E.M. ** *p* < 0.01 vs. Ctrl; *** *p* < 0.001 vs. Positive Ctrl; **** *p* < 0.0001 vs. Positive Ctrl; °° *p* < 0.01 vs. Ica 1 μM; ^^ *p* < 0.01 vs. Ica 5 μM; # *p* < 0.05 vs. Doxo; Tukey’s multiple comparisons test and Mann Whitney test (*n* = 3). Data is for three independent experiments.

**Figure 9 nutrients-13-04070-f009:**
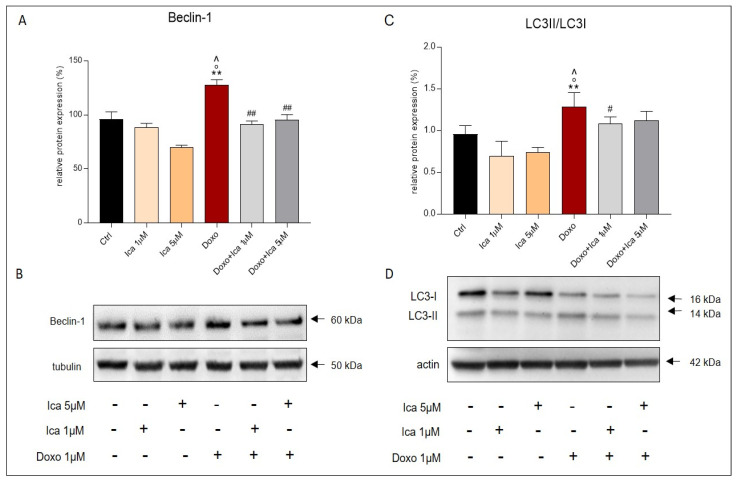
The effect of Ica on the pro-autophagic markers Beclin-1 and LC3. Treatment with Doxo alone significantly increased the expression levels of Beclin-1 (**A**) and LC3II/LC3I ratio (**C**) in H9c2 cells. Pretreatment with Ica at both concentrations used inhibited the proautophagic pathway when compared to H9c2 cells treated with Doxo alone. (**B**,**D**) Representative images of western blot analysis. Three independent Western Blots were quantified by densitometry. The protein expression of Beclin-1 and LC3II/LC3I were normalized with respect to the corresponding tubulin or actin signals of the appropriate samples and expressed as a percentage. Data are expressed as mean ± S.E.M. ** *p* < 0.01 vs. Ctrl; ° *p* < 0,05 vs. Ica 1uM; ^ *p* < 0.05 vs. Ica 5 μM; # *p* < 0.05 vs. Doxo; ## *p* < 0.01 vs. Doxo; Mann Whitney test (*n* = 3).

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
