# Peer review of "Icariin Protects H9c2 Rat Cardiomyoblasts from Doxorubicin-Induced Cardiotoxicity: Role of Caveolin-1 Upregulation and Enhanced Autophagic Response"

_nutrients, 2021, doi:10.3390/nu13114070_

Round 1
Reviewer 1 Report
Below are comments and recommendations to improve the study:
- The introduction is too long. Needs to be rewritten succinctly and be limited to the investigative mechanisms
- Figure1. Doxo shows 10% decrease in cell death but in figure 2 it shows 40% decrease in cell death. The authors need to explain this difference.
- The authors may consider representing the graphs with the control taken as 100% of cell viability and compare all other conditions to 100% of cell viability.
- Figure 2 B. should include the Ica 1 uM and 5 um conditions on the same graph to be able to compare all conditions to each others.
- lane 355: what does the last sentence mean?
- figure 3A, 7A, 9A and 9C. What do you mean by "%" on the y-axis; the % of what?
- The discussion section has to be rewritten succinctly and refocused on the thesis of the manuscript.
Reviewer 2 Report
In the manuscript #nutrients-1443868, by Scicchitano et al. "Icariin protects H9c2 rat cardiomyoblasts from doxorubicin-induced cardiotoxicity: role of Caveolin-1 upregulation and enhanced autophagic response." the authors showed the protective activities of Icariin (Ica) against Doxo-detrimental effects on rat heart-tissue derived embryonic cardiac myoblasts through the reduction in Caveolin-1 expression and the inhibition of phosphodiesterase 5 (PDE5a) activity. This manuscript seems to lack sufficient novelty for readers because these main observations were already reported in the previous papers (1 -5). The additional data concerning Ica in molecular mechanisms which are included in this manuscript add little to what is already known. Although I have no concern about the quality of the presented data which can be confirmed, Fig. 2 and 4 are broken in the process of conversion from the manuscript to pdf.
References
The cardioprotective effects:
1) Zhang L, Wang S, Li Y, Wang Y, Dong C, Xu H. Cardioprotective effect of icariin against myocardial fibrosis and its molecular mechanism in diabetic cardiomyopathy based on network pharmacology: Role of ICA in DCM. Phytomedicine. 2021 Oct;91:153607. doi: 10.1016/j.phymed.2021.153607. PMID: 34411833
2) Sharma S, Khan V, Dhyani N, Najmi AK, Haque SE. Icariin attenuates isoproterenol-induced cardiac toxicity in Wistar rats via modulating cGMP level and NF-kappaB signaling cascade. Hum Exp Toxicol. 2020 Feb;39(2):117-126. doi: 10.1177/0960327119890826. PMID: 31797691
The reduction in Caveolin-1 expression:
3) Liu QW, Yang ZH, Jiang J, Jiang R. Icariin modulates eNOS activity via effect on post-translational protein-protein interactions to improve erectile function of spontaneously hypertensive rats. Andrology. 2021 Jan;9(1):342-351. doi: 10.1111/andr.12875.PMID: 33507631
The inhibition of phosphodiesterase 5:
4) Dell'Agli M, Galli GV, Dal Cero E, Belluti F, Matera R, Zironi E, Pagliuca G, Bosisio E. Potent inhibition of human phosphodiesterase-5 by icariin derivatives. J Nat Prod. 2008 Sep;71(9):1513-7. doi: 10.1021/np800049y. PMID: 18778098
5) Xin ZC, Kim EK, Lin CS, Liu WJ, Tian L, Yuan YM, Fu J. Effects of icariin on cGMP-specific PDE5 and cAMP-specific PDE4 activities. Asian J Androl. 2003 Mar;5(1):15-8.
Round 2
Reviewer 2 Report
I have again enjoyed reviewing your manuscript according to the Author Response, which is quite convincing. I understood this study provides an important contribution to a promising therapeutic approach to attenuate cardiac toxicity induced by doxorubicin.